# COVID-19 Pandemic Brings a Sedentary Lifestyle in Young Adults: A Cross-Sectional and Longitudinal Study

**DOI:** 10.3390/ijerph17176035

**Published:** 2020-08-19

**Authors:** Chen Zheng, Wendy Yajun Huang, Sinead Sheridan, Cindy Hui-Ping Sit, Xiang-Ke Chen, Stephen Heung-Sang Wong

**Affiliations:** 1Department of Sports Science and Physical Education, The Chinese University of Hong Kong, Hong Kong 00852, China; zhengchen@link.cuhk.edu.hk (C.Z.); sineadsheridan@cuhk.edu.hk (S.S.); sithp@cuhk.edu.hk (C.H.-P.S.); 2Department of Sport and Physical Education, Hong Kong Baptist University, Hong Kong 00852, China; wendyhuang@hkbu.edu.hk; 3School of Biomedical Sciences, The University of Hong Kong, Hong Kong 00852, China; xkchen@hku.hk

**Keywords:** COVID-19, physical activity, sedentary behavior, sleep, young adults

## Abstract

The coronavirus disease 2019 (COVID-19) pandemic continues to pose profound challenges to society. Its spread has been mitigated through strategies including social distancing; however, this may result in the adoption of a sedentary lifestyle. This study aimed to investigate: (1) physical activity (PA) levels, sedentary behavior (SB), and sleep in young adults during the COVID-19 epidemic, and (2) the change in these behaviors before and during the pandemic. A total of 631 young adults (38.8% males) aged between 18 and 35 participated in the cross-sectional study and completed a one-off online survey relating to general information, PA, SB, and sleep. For the longitudinal study, PA, SB, and sleep data, obtained from 70 participants before and during the COVID-19 pandemic, were analyzed. Participants engaged in low PA, high SB, and long sleep duration during the COVID-19 pandemic. Moreover, a significant decline in PA while an increase in time spent in both SB and sleep was observed during the COVID-19 outbreak. The results of this study demonstrated a sedentary lifestyle in young adults during the COVID-19 pandemic, which will assist health policymakers and practitioners in the development of population specific health education and behavior interventions during this pandemic and for other future events.

## 1. Introduction

Since the first outbreak of the coronavirus disease 2019 (COVID-19) in Wuhan, China, in early December 2019 [1], the disease has rapidly spread across the world [2]. The first confirmed case in Hong Kong was reported in late January 2020 [3], and an escalation in the number of cases was observed in late March and early July 2020, respectively [4]. According to the World Health Organization (WHO), recently, this pandemic has infected more than 20 million people from over 200 countries around the world and has resulted in over 730,000 deaths [5]. Thus, unprecedented efforts have been made by Governments across the globe to contain the epidemic—e.g., quarantine, social distancing, and the isolation of infected individuals [6]. The efforts made by the Hong Kong Government, for instance, included border entry restrictions, quarantine and the isolation of cases and contacts, and the closure of schools, resulting in major disruptions to daily routines [7]. An escalation in the number of cases in Hong Kong in late March 2020 further fueled the Government to enforce stricter measures, including the closure of leisure facilities and cultural facilities [8], and the continued delivery of courses to students via online platforms for the remainder of the academic term.

While these measures are highly commendable and critical to mitigate the spread of COVID-19, they may result in inducing unhealthy behaviors, such as a sedentary lifestyle, with most individuals adhering to social distancing by working or studying from home or, in other cases, self-isolating under strict quarantine. Physical activity (PA), sedentary behaviors (SB), and sleep are three behaviors that occupy a large proportion of an individual’s time over 24 h. Under normal circumstances, a sedentary lifestyle, including physical inactivity and prolonged SB, has been previously identified as problematic among adults globally, with one-third of adults physically inactive and 41.5% spending four or more hours per day sitting [9]. An inactive and sedentary lifestyle are closely related to a higher risk for cardiovascular diseases [10]. In addition, short sleep duration during the weekday, late sleep times, and a variable sleep schedule are all associated with cardiometabolic health and weight gain [11,12].

In early March, researchers anticipated that social distancing, including the closure of schools and home confinement, may result in less PA, prolonged SB and poor sleep quality [13]. To date, several studies have reported relevant data for different populations in various countries. A longitudinal study from Shanghai, China found that children and adolescents engaged in 435 min less PA and spent 28 more hours in screen time per week since the COVID-19 outbreak [14]. A national survey in Canada also found that children and youths had lower PA levels, higher SB (including leisure screen time), and more sleep during the COVID-19 pandemic [15]. Similar results have been reported for adults, such as a negative change in PA and sleep in Australia [16], and 40.5% of inactive Canadians becoming less active [17]. Moreover, a national survey, including 35 research organizations, reported a negative effect of COVID-19 home confinement on all PA intensity levels and an increase in daily sitting time [18].

Accordingly, there is lack of evidence for the influence of the COVD-19 pandemic on lifestyle behaviors in young adults, particularly from Asia. Thus, this study aimed to investigate: (1) PA levels, time spent in SB, and sleep in Hong Kong young adults during the COVID-19 pandemic; (2) the changes in these lifestyle behaviors after the COVID-19 outbreak.

## 2. Materials and Methods

### 2.1. Study Design and Participants

The study design included both a cross-sectional and longitudinal study. For recruitment for the cross-sectional study, information was advertised online and through word of mouth. The inclusion criteria for study participation included: (1) general healthy adults aged 18–35 years old and (2) living in Hong Kong for the past two months. Participants completed an English online survey supported by Google form (Google LLC, Mountain View, CA, USA), which included five components: general information (e.g., age, sex, body weight, and height), COVID-19 related issues, PA, SB, and sleep. Participant body mass index (BMI) was subsequently calculated as weight in kilograms divided by height in meters squared. The online survey was conducted between 15 April 2020 and 26 April 2020, with a total of 631 respondents. For the longitudinal study, 70 young adults’ baseline data—from a previous cross-sectional study conducted in 2019 by our research group that investigated the relationship between SB and risks of cardiovascular disease (unpublished data)—were obtained. The 70 adults were invited to report their lifestyle behaviors during the pandemic. Of those participants, 60 completed all three questionnaires to assess their PA, SB, and sleep, while only 10 participants completed the PA questionnaire. The same questionnaires were used at baseline and for the follow-up measurement. The cross-sectional study has similar characteristics to the longitudinal study. The protocol was approved by the Survey and Behavioral Research Ethics, The Chinese University of Hong Kong (SBRE-19-599).

### 2.2. Physical Activity

The International Physical Activity Questionnaires (IPAQ) was used to assess the PA level in participants. The short version of IPAQ’s validity and reliability have been tested in 12 countries [19], which has been shown to be suitable for population surveillance and large-scale studies. Three items, vigorous PA (VPA), moderate PA (MPA), and walking, were assessed by IPAQ. Moderate to vigorous PA (MVPA) was calculated by adding MPA and VPA. The MET-minutes per week (MET.min/week) were calculated using the following formula: intensity (MET) × duration × frequency. In addition, to assess the impact of COVID-19 on PA, one more question was asked: “How has your physical activity levels been since the onset of the COVID-19 pandemic? (e.g., increase, no change, and decrease)”.

### 2.3. Sedentary Behavior

SB was measured using the Sedentary Behavior Questionnaire (SBQ) in participants, which has been previously validated in adults [20]. Intra-class correlation coefficients for all nine items and total scale were acceptable (range = 0.51–0.93) [20]. A total of nine SBs (TV/DVD, computer/video games, sitting listening to music, sitting talking on telephone, doing computer/paper work, reading books, playing musical instrument, doing arts and crafts, and sitting for transport) were selected for this questionnaire. All items were assessed for a usual weekday and weekend day for the past month with nine options: none, ≤15 min, 30 min, 1 h, 2 h, 3 h, 4 h, 5 h, and ≥6 h. Based on the previous methodology published, the time spent on each behavior was converted into hours (e.g., a response of 15 min was recorded as 0.25 h) [20]. To obtain daily estimates, each item of weekday hours was multiplied by 5 and weekend hours were multiplied by 2, and these were then divided by 7. The daily SB was assessed by IPAQ with a separate question asking about sitting time [20].

### 2.4. Sleep

The most commonly subjective sleep scale, the Pittsburgh Sleep Quality Index (PSQI), was used to assess both sleep quality and sleep duration in participants. The PSQI is a validated 19-item, self-reported questionnaire, which is categorized into seven sleep quality components (subjective sleep quality, sleep latency, sleep duration, habitual sleep efficiency, sleep disturbances, use of sleeping medication, and daytime dysfunction) [21]. The final score of the seven sleep components ranged from 0 to 21 points. Final scores of 5 or <5 points are classified as having “good sleep quality”, and >5 points are classified as “poor sleep quality” [21]. The sleep duration was calculated from participants’ reported bed and wake-up times. Besides, to assess the impact of COVID-19 on sleep quality, one more question was asked: “How was your sleep since the onset of the COVID-19 pandemic? (e.g., better than usual, the same as usual, and worse than usual)”.

### 2.5. COVID-19 Related Issues

Participants were also asked the following five questions related to COVID-19: (1) “Please identify your main source of information regarding the COVID-19 pandemic: Newspapers or Television, Government websites, Work colleagues/friends, Facebook/Twitter/Instagram/YouTube”; (2) “Have you ever been home quarantined or stayed in a quarantine center for compulsory quarantine? Yes, No or Prefer not to say”; (3) “I am concerned about contracting COVID-19 myself”; (4) “I am concerned about other family members or friends contracting COVID-19”—answers from participants for questions (3) and (4) used one of the following five options: “not at all concerned, slightly concerned, somewhat concerned, moderately concerned, extremely concerned”; (5) “How often do you practice these prevention strategies against the spread of COVID-19?”—for this question, three of the most common and effective prevention strategy methods were chosen, including “regular hand-washing with soap, wearing a face mask, and avoiding restaurants/gyms/shops”. All three items were answered using the following five options: “always, often, sometimes, rarely, and never”.

### 2.6. Statistical Analysis

Three international guidelines for PA, SB and sleep for adults for health were applied for data analysis: (1) achievement of at least 150 min of moderate-intensity aerobic PA or at least 75 min of vigorous-intensity aerobic PA throughout the week [22]; (2) engagement in <9 h of SB per day for adults [23]; (3) score of sleep quality <5 with sleep duration between 7 and 9 h [24]. Descriptive information (all and stratified by sex), including participant characteristics, COVID-19 related issues, and participants’ daily behaviors, were summarized and reported as means ± standard deviation (SD) or median (interquartile range) for continuous variables and as proportions of participants for categorical variables. The normality analysis was applied to all variables. Because VPA, MPA, MVPA, walking, and MET min/week are not a normal distribution, the Wilcoxon Signed-Rank Test was used to assess the differences between males and females for the above variables. For other variables, independent samples *t*-tests and Chi-Square tests were used for continuous variables and categorical variables, respectively. The change in participants’ daily behaviors (e.g., PA, SB, and sleep) was determined using paired sample t-tests and shown as means ± SD. All statistical tests were performed using SPSS for Windows, version 24 (IBM Corp., Armonk, NY, USA).

## 3. Results

### 3.1. Descriptive Statistics of Participant

A total of 631 participants (38.8% males) were included in data analysis. The characteristics of the participants are shown in Table 1. The participants’ mean age was 21.1 years—specifically, 67.1% of them younger than 22 years, 23.5% of them between 22 and 25 years, and 9.4% of them older than 25 years. Based on the WHO’s recommendations for Asian adults [25], 12.2% and 23.3% of the males and females were overweight, respectively.

### 3.2. Lifestyle Behavior during COVID-19 Pandemic

Lifestyle behaviors (e.g., PA, SB, and sleep) are presented in Table 2, where 30% of participants met the PA guideline, and more than half of the participants (57.8%) did not engage in any VPA during the COVID-19 pandemic. In total, 70% of participants reported that their PA level decreased since the onset of the COVID-19 pandemic. Compared to males, engagement in computer/video games was lower while engagement in computer/paper work and arts and crafts was higher in females. Despite the fact that females had a significantly longer mean sleep duration than males (8.7 ± 1.2 vs. 8.5 ± 1.2, *p* < 0.05), more male participants met the sleep guidelines than females (46.5 vs. 38.1, *p* < 0.05).

### 3.3. Changes in Lifestyle Behaviors after COVID-19 Outbreak

The changes in lifestyle behaviors in participants from the longitudinal study are presented in Figure 1. After the COVID-19 outbreak, all PA levels, including VPA (before vs. during COVID-19, 9.5 ± 12.5 vs. 6.0 ± 11.6, *p* < 0.05), MPA (11.2 ± 16.0 vs. 5.5 ± 8.7, *p* < 0.01), and walking (39.7 ± 30.7 vs. 19.8 ± 24.5, *p* < 0.01), were significantly declined, while both time spent in SB (7.8 ± 3.2 vs. 10.0 ± 3.2, *p* < 0.01) and sleep duration (7.7 ± 1.0 vs. 8.4 ± 1.2, *p* < 0.01) significantly increased. A lower percentage of participants met the guidelines for PA (50.0% vs. 20.0%), SB (54.3% vs. 40.0%), and sleep (67.1% vs. 57.1%) after the COVID-19 outbreak. When considering the SB by types of activities, engagement in TV/DVD (0.9 ± 0.8 vs. 1.7 ± 1.4, *p* < 0.01) and computer/paper work (2.2 ± 1.7 vs. 3.1 ± 2.0, *p* < 0.01) was significantly higher and sitting time during transportation (0.7 ± 0.7 vs. 0.4 ± 0.6, *p* < 0.01) was significantly lower during COVID-19 than before the epidemic.

### 3.4. COVID-19 Related Issues

COVID-19 related issues are presented in Table 3, including the main sources of information, concern of contracting COVID-19, and prevention strategies. Generally, females had greater concern for contracting COVID-19 themselves and for their family members contracting COVID-19. Therefore, females engaged more in COVID-19 prevention strategies, such as wearing a face mask and avoiding restaurants, gyms, and shops, compared to males.

## 4. Discussion

This is the first study, to the best of our knowledge, to investigate PA levels, time spent in SB, and sleep in young adults during the COVID-19 pandemic and the changes in these lifestyle behaviors after the COVID-19 outbreak in Hong Kong. The major findings of our study were that engagement in all PA behaviors significantly declined while time spent in SB and sleep duration significantly increased following the COVID-19 outbreak.

Our finding of a decrease in all types of PA (i.e., MPA, VPA and walking) after the COVID-19 outbreak are consistent with a recent national survey in Canada, which reported a significant decline in all physical activities in children and adolescents [15]. The cross-sectional analysis also revealed the low volume of PA that participants engaged in during the COVID-19 pandemic, with an average of 3 min/day spent in MPA and 17 min/day walking. Similarly, Xiang et al. reported that children and adolescents engaged in 105 min per week during the COVID-19 pandemic in Shanghai [14]. The low volume of PA undertaken by participants may be due to social distancing (e.g., cancellation of all team sports training and competitions and the closure of public leisure facilities and gymnasiums), working from home and concern of the threat posed by COVID-19 in Hong Kong. Furthermore, the limited living space in Hong Kong, which is smaller than in other Asian cities, including Tokyo and Singapore, may further restrict the opportunities for young adults to exercise at home [26].

The total time spent in SB during the waking day was significantly higher during COVID-19 than prior to the outbreak. This is consistent with a national survey, which reported that daily sitting time increased from 5 to 8 h (28.6%) per day during home confinement [18]. Similarly, children and adolescents in both Shanghai and Canada engaged in more than 5 h of screen time per day [14,15]. This may be partially explained by young adults engaging in social distancing by staying home, and online teaching, which subsequently resulted in prolonged screen time, such as elevated time spent watching TV, playing computer games, and online teaching [13]. Specifically, students spent less time on transportation, less time in P.E. class, and the majority of the academic term in screen time during the online teaching period. Consequently, we also found that time spent in both TV/DVD and computer/paper work significantly increased while sitting time for transport decreased following the COVID-19 outbreak.

Increased screen time, which was reported in this study, has been previously shown to be accompanied by prolonged sedentary bouts without interruption, which have a more negative impact on health outcomes [27]. Interventions that increase PA while decrease SB in young adults during the COVID-19 pandemic are warranted, especially for inactive individuals as they are more likely to become less active during the COVID-19 pandemic [17]. For instance, simple home-based exercises, such as body resistance training or high-intensity interval training (HIIT), can be applied to this population. Appropriate health education and behavior interventions via online platforms can also be developed during this pandemic, as more than 70% of participants reported that their engagement in PA had significantly decreased during the COVID-19 pandemic. Interestingly, one study reported that women who spent more time exercising outdoors had better mental and general health than those who were not [28]. Moreover, maintaining regular PA during the COVID-19 pandemic is important for preventing future chronic health conditions due to a sedentary lifestyle [29].

The COVID-19 pandemic brings significant disruptions to daily routines. With a more flexible schedule due to the closure of schools, colleges, universities and businesses, participants had significantly longer sleep duration during the pandemic. Of particular interest is that 37% of participants reported having poorer sleep quality during the COVID-19 pandemic. Since poor sleep is highly correlated with stress [30], poor sleep quality reported by participants may be due to the stress induced by the threat of COVID-19. Notably, 40.7% of Australian adults reported a negative change in sleep since the onset of COVID-19 because of change in exercise behaviors, and employment and relationship concerns [16]. Thus, maintaining a regular sleep routine during the COVID-19 pandemic is essential. Furthermore, the effect of COVID-19 on sleep and mental health among young adults warrants further investigation.

Individual behaviors of young adults have been changed in response to the threat posed by COVID-19. Based on our results, 90% of participants reported that they always wear a face mask when leaving home, and only 0.3% of participants reported never wearing a face mask. Previous research has shown that wearing face-masks during exercise causes a significant increase in physiological demand [31]. This may influence PA or exercise behaviors of individuals. Similarly, a recent study found that 99% of participants reported wearing face masks when leaving home [7]. In addition, 85% of participants reported that they always or often wash their hands with soap. When the severe acute respiratory syndrome (SARS) occurred in Hong Kong in 2003, the proportion of face mask use and washing hands among adults was 79% and 82%, respectively [32]. Interestingly, females showed more concerns about contracting COVID-19 themselves or their family members contracting it, and therefore engaged in more prevention strategies that included always wearing a face mask and avoiding restaurants/gyms/shops than males, which may lead to gender-difference in reduced outdoor activities during the pandemic.

One major strength of this study is that both cross-sectional and longitudinal analyses were applied. Secondly, there was a relatively large sample size in the cross-sectional analysis. Thirdly, three behaviors (PA, SB, and sleep), which occupy a large proportion of time in individuals over 24 h were assessed in the current study.

The limitations of this study include the use of subjective measurements to assess PA, SB and sleep, which are associated with increased risk of bias. Though all the questionnaires used in this study have been previously validated, objective measurement, such as the use of an accelerometer, would be more accurate in assessing PA and SB in participants. Moreover, the longitudinal study has a limited sample size. Finally, the current study may have selection bias, as participation was voluntary.

## 5. Conclusions

Low PA levels, high amount of time spent on SBs, and long sleep duration were identified in young adults during the COVID-19 pandemic, with less than half of the participants meeting any of the recommended guidelines for PA, SB or sleep. There was also a significant reduction in PA behaviors and a significant increase in SB and the sleep duration of young adults following the outbreak of COVID-19. These findings may have important public health implications and provide evidence for future intervention studies.

## Figures and Tables

**Figure 1 ijerph-17-06035-f001:**
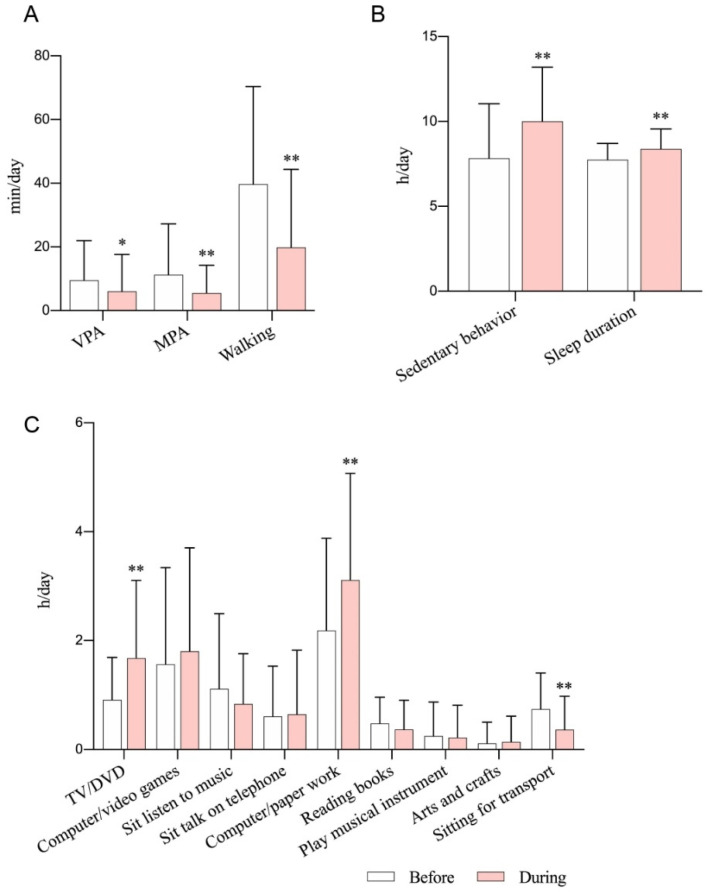
The changes in participants’ lifestyle behaviors. (**A**) Physical activity, (**B**) sedentary behavior and sleep duration, (**C**) sedentary behavior by types by activities. MPA: moderate physical activity; VPA: vigorous physical activity. * *p* < 0.05, ** *p* < 0.01, compared with before.

**Table 1 ijerph-17-06035-t001:** Participant characteristics for cross-sectional study and stratified by sex.

Variables	Mean ± SD			
	All (*n* = 631)	Males (*n* = 245)	Females (*n* = 386)	*p* Value
Age (years)	21.1 ± 2.9	21.5 ± 3.2	20.9 ± 2.5	0.01
Height (cm)	165.6 ± 8.3	173.2 ± 6.1	160.8 ± 5.5	<0.01
Weight (kg)	57.0 ± 10.1	64.3 ± 10.0	52.4 ± 7.0	<0.01
BMI (kg/m^2^)	20.7 ± 2.6	21.4 ± 2.9	20.3 ± 2.4	<0.01

BMI: body mass index; SD: standard deviation

**Table 2 ijerph-17-06035-t002:** Participants’ lifestyle behaviors for cross-sectional study and stratified by sex.

Variables	Median (IQR) or Mean ± SD or %			
	All (*n* = 631)	Males (*n* = 245)	Females (*n* = 386)	*p* Value
PA	-	-	-	-
Walking (min/day)	17.1 (28.6)	17.1 (27.1)	12.85 (28.6)	0.07
Moderate PA (min/day)	2.9 (11.4)	2.85 (12.9)	2.85 (10.4)	0.30
Vigorous PA (min/day)	0.0 (8.6)	0.0 (10.0)	0.0 (8.6)	0.16
Moderate to Vigorous PA(min/day)	8.6 (25.0)	8.6 (25.7)	5.7 (20.7)	0.18
Total Energy Expenditure (MET min/week)	792.0 (1398.0)	864.0 (1836.0)	792.0 (1230.0)	0.07
Change of PA Level	-	-	-	0.05
Increase	16.5	12.2	19.2	-
No Change	11.3	10.6	11.7	-
Decrease	72.3	77.1	69.2	-
Meet PA Guideline ^a^	29.6	30.2	29.3	0.80
SB	-	-	-	-
TV/DVD (h/day)	1.7 ±1.5	1.6 ± 1.5	1.8 ± 1.5	0.26
Computer/Video Games (h/day)	1.4 ±1.6	1.9 ± 1.8	1.1 ± 1.4	<0.01
Sitting Listening to Music (h/day)	1.1 ± 1.3	1.1 ± 1.4	1.1 ± 1.3	0.65
Sitting Talking on Telephone (h/day)	0.8 ± 1.2	0.9 ± 1.3	0.8 ± 1.2	0.90
Computer/Paper Work (h/day)	3.1 ± 1.8	2.9 ± 1.8	3.3 ± 1.8	<0.01
Reading Books (h/day)	0.5 ± 0.9	0.5 ± 0.9	0.6 ± 0.9	0.39
Play Musical Instrument (h/day)	0.2 ± 0.5	0.2 ± 0.4	0.2 ± 0.6	0.52
Doing Arts and Crafts (h/day)	0.2 ± 0.5	0.1 ± 0.4	0.2 ± 0.6	<0.01
Sitting for Transport (h/day)	0.4 ± 0.6	0.4 ± 0.7	0.4 ± 0.5	0.15
Daily SB (h/day)	9.4 ± 3.0	9.4 ± 3.2	9.5 ± 3.0	0.77
Meet SB Guideline ^b^	42.5	44.1	41.5	0.97
Sleep	-	-	-	-
Sleep Duration (h/day)	8.6 ± 1.2	8.5 ±1.2	8.7 ± 1.2	0.02
Sleep Quality	5.2 ± 2.5	4.9 ± 2.3	5.3 ± 2.6	0.07
Meet Sleep Guideline ^c^	41.4	46.5	38.1	0.04
Change in Sleep Quality	-	-	-	0.07
Better than Usual	18.4	20.8	16.8	-
The Same as Usual	43.9	46.9	42.0	-
Worse than Usual	37.7	32.2	41.2	-

IQR: interquartile range; PA: physical activity; SB: sedentary behavior; SD: standard deviation. ^a^ At least 150 min of moderate-intensity aerobic PA or at least 75 min of vigorous-intensity aerobic PA throughout the week. ^b^ Less than 9 h of SB per day. ^c^ The score of sleep quality <5 with the sleep duration in 7–9 h.

**Table 3 ijerph-17-06035-t003:** COVID-19 related issues for cross-sectional study and stratified by sex.

Variables	%			
	All (*n* = 631)	Males (*n* = 245)	Females (*n* = 386)	*p* Value
Main Sources of Information	-	-	-	0.12
Newspapers or TV	37.4	33.5	39.9	-
Government Websites	2.4	1.2	3.1	-
Work Colleagues/Friends	1.0	0.8	1.0	-
Facebook/Twitter/Instagram/YouTube	59.3	64.5	56.0	-
Home Quarantine/Compulsory Quarantine	-	-	-	0.35
Yes	7.1	5.7	8.0	-
No	92.1	93.1	91.5	-
Prefer not to Say	0.8	1.2	0.5	-
Contracting COVID-19 Myself	-	-	-	<0.01
Not at All Concerned	3.2	5.7	1.6	-
Slightly Concerned	13.6	15.5	12.4	-
Somewhat Concerned	22.5	25.3	20.7	-
Moderately Concerned	43.1	37.6	46.6	-
Extremely Concerned	17.6	15.9	18.7	-
Family Members Contracting COVID-19	-	-	-	0.02
Not at all Concerned	1.4	2.4	0.8	-
Slightly Concerned	9.0	12.2	7.0	-
Somewhat Concerned	20.6	23.3	18.9	-
Moderately Concerned	43.4	38.4	46.6	-
Extremely Concerned	25.5	23.7	26.7	-
Prevention Strategies	-	-	-	-
Hand-Washing with Soap	-	-	-	0.07
Always	55.6	49.8	59.3	-
Often	32.5	35.5	30.6	-
Sometimes	9.4	11.0	8.3	-
Rarely	2.2	2.9	1.8	-
Never	0.3	0.8	0.0	-
Wearing a face mask	-	-	-	0.02
Always	88.9	84.9	91.5	-
Often	8.1	9.4	7.3	-
Sometimes	2.1	3.7	1.0	-
Rarely	0.5	0.8	0.3	-
Never	0.5	1.2	0.0	-
Avoiding Restaurants/Gyms/Shops	-	-	-	0.02
Always	35.0	28.6	39.1	-
Often	34.2	34.7	33.9	-
Sometimes	24.4	27.3	22.5	-
Rarely	5.5	8.2	3.9	-
Never	0.8	1.2	0.5	-

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
