# Peer review of "COVID-19 Pandemic Brings a Sedentary Lifestyle in Young Adults: A Cross-Sectional and Longitudinal Study"

_ijerph, 2020, doi:10.3390/ijerph17176035_

Round 1

Reviewer 1 Report

The manuscript "COVID-19 Pandemic Brings a Sedentary Lifestyle in 2 Young Adults: a Cross-Sectional and Longitudinal 3 Study" is another effort to understand how this pandemic is affecting global health and initiatives are welcomed to collaborate with the current knowledge about changes in behavior.

Overall I am not sure how the subsample (n=70) is actually improving the manuscript or making it more complex to grasp, for example, when the authors write about changes, without having evaluated previous behavior in the large sample, it is kind of misleading.

Specific suggestions.
Line 69 - along all the methods, revise the use of the word "trial".
Line 77 - what do the authors mean with "our previous research"? Where did these 70 subjects come from? It must be explained, especially because there is no reference to help the reader to discover anything about this subsample.
Line 104 - I believe it is with instead of which.
Line 225 - I did not understand why online teaching would increase sedentary time, as time spent in these lessons would anyway be spent in school, sitting if we were not during pandemics.
Line - 248 - one of the major limitations of this study is the selection bias, as participation was voluntary

Reviewer 2 Report

The article addresses an interesting and important issue due to the global pandemic we are experiencing. Here are some errors and suggestions for improvement:

  • line 20: it would be necessary to specify the type of survey. Specify if it was Ad-hoc or built from instruments to be validated.
  • As the manuscript is oriented towards the young adult population, it would be recommended that the introduction to the article speaks more about this population specifically.
  • line 77: researchers should explain what the previous study they conducted was about, to see if it is related to the current one. It should be specified whether the questionnaire was the same on both occasions.
  • It is certain that already validated questionnaires have been chosen, but all of them have been modified, selecting only some items or adding others. I think that the authors should do at least one reliability analysis (Cronbach's Alpha).
  • Authors should start the statistical analysis by performing a normality analysis, to see what type of sample it is and to find out whether they should use parametric or non-parametric tests.
  • I don´t understand the reason for Table 2, since knowing that information is not part of the study aim.
  • In table 1, where do the statistically significant differences come from? Significance values do not appear in the table. The same is applicable for the other tables. The value of "p" must be added.

In general, I recommend that the authors focus on the methodology and results section, which are the sections with the most deficiencies. Statistical analysis is very poor, and not up to a journal like this. The objective of the study is not clear and it seems like two works within one (one longitudinal and one transversal). Perhaps they should be separated into two independent studies.

Reviewer 3 Report

REVIEW

Manuscript number ijerph-885338

Title: COVID-19 pandemic brings a sedentary lifestyle in young adults: a cross-sectional and longitudinal study

GENERAL COMMENTS:

The manuscript addresses important issue of sedentary lifestyle in young adults as a result of COVID-19 pandemic. Healthy lifestyle has a big impact on the immunity system and the ability of the body to fight any disease. It is really important to conduct and publish research in this topic.

I have some additional comments and suggestions:

Abstract

As stated in the instructions for authors ‘The abstract should be a single paragraph and should follow the style of structured abstracts, but without headings’. Please correct.

Introduction

I would suggest to add in the introduction the most recent publication in the topic as the subject of COVID-19 pandemic is very recent.

Examples of the papers published in the topic:

- Ammar A, Brach M, Trabelsi K, et al. Effects of COVID-19 Home Confinement on Eating Behaviour and Physical Activity: Results of the ECLB-COVID19 International Online Survey. Nutrients. 2020;12(6):1583. Published 2020 May 28. doi:10.3390/nu12061583

- Lesser IA, Nienhuis CP. The Impact of COVID-19 on Physical Activity Behavior and Well-Being of Canadians. Int J Environ Res Public Health. 2020;17(11):3899. Published 2020 May 31. doi:10.3390/ijerph17113899

- Stanton R, To QG, Khalesi S, et al. Depression, Anxiety and Stress during COVID-19: Associations with Changes in Physical Activity, Sleep, Tobacco and Alcohol Use in Australian Adults. Int J Environ Res Public Health. 2020;17(11):4065. Published 2020 Jun 7. doi:10.3390/ijerph17114065

Materials and Methods

Page 2, Lines 77-78

The authors mention study conducted in 2019 from which they used the data for the longitudinal study. There should be more details about this study. Were the results published?

Were the questionnaires used in the study translated into Chinese?

Discussion

I would suggest to expand the discussion with the most current literature for example:

- Ammar A, Brach M, Trabelsi K, et al. Effects of COVID-19 Home Confinement on Eating Behaviour and Physical Activity: Results of the ECLB-COVID19 International Online Survey. Nutrients. 2020;12(6):1583. Published 2020 May 28. doi:10.3390/nu12061583

- Lesser IA, Nienhuis CP. The Impact of COVID-19 on Physical Activity Behavior and Well-Being of Canadians. Int J Environ Res Public Health. 2020;17(11):3899. Published 2020 May 31. doi:10.3390/ijerph17113899

- Stanton R, To QG, Khalesi S, et al. Depression, Anxiety and Stress during COVID-19: Associations with Changes in Physical Activity, Sleep, Tobacco and Alcohol Use in Australian Adults. Int J Environ Res Public Health. 2020;17(11):4065. Published 2020 Jun 7. doi:10.3390/ijerph17114065

Minor comments

  • There are some spelling mistakes in the text and the whole paper needs to be checked and corrected by native speaker
  • The list of references should be prepared according to the instructions for authors. For example references 16 and 20 have to be corrected.

Reviewer 4 Report

Thank you very much for the opportunity to review the work entitled “COVID-19 Pandemic Brings a Sedentary Lifestyle in Young Adults: a Cross-Sectional and Longitudinal Study”. I believe that it contributes to objectify a significant problem derived from the pandemic.

  1. The aim of this study was to investigate the PA levels, sedentary behaviour and sleep among Young adults. In the introduction is covered adequately the PA levels and SB, however, the sleep is almost no covered. Please add a paragraph about sleep to understand why was assessed.
  2. Do you exclude persons with musculoskeletal diseases?
  3. One of the most critical parameters reported in PA studies is the MVPA; please report in the text and the tables.
  4. One of the inclusion criteria is 18 to 35 years old. However, the mean age was 21.1 (2.9) years. This means that the number of young people under 23 years of age was very high. Physical activity varies between people who study and people who work. What percentage of the sample was studying? How many worked?
  5. Could the socio-political situation in Hong Kong have influenced the level of physical activity? I am referring to the interruption of activities such as free movement to the University (since the sample is young)
  6. The authors report that the use of face masks could contribute to a decrease in physical activity. In my opinion, that is a hazardous statement. The authors did not assess the barriers and facilitators of physical activity. On the other hand, more influence has the policies of the different governments in promoting that people stay at home together with all the restrictive measures that this entails.
  7. In the second paragraph of the discussion, after the ruling on the masks, the authors refer to handwashing with soap and water. Also, the authors contrast the data with SARS. In this paragraph, the possible causes of the decrease in outdoor activities are being discussed, but I do not see the relationship of washing hands with soap and water, considering that this is not sedentary behaviour. The information is undoubtedly very important, however, I believe that it should be ordered from the central aspects of the investigation and later in the discussion, refer to hand washing.

Round 2

Reviewer 2 Report

After reviewing the manuscript I see that the authors have modified most of the suggestions I made to them in the last review, so I now consider it appropriate for publication.

Reviewer 4 Report

My impression is that the authors have corrected the reviewers' suggestions. The text has improved considerably.